# Neural Correlates of Delay Discounting in the Light of Brain Imaging and Non-Invasive Brain Stimulation: What We Know and What Is Missed

**DOI:** 10.3390/brainsci13030403

**Published:** 2023-02-26

**Authors:** Andrea Stefano Moro, Daniele Saccenti, Mattia Ferro, Simona Scaini, Antonio Malgaroli, Jacopo Lamanna

**Affiliations:** 1Department of Psychology, Sigmund Freud University of Milan, 20143 Milan, Italy; 2Center for Behavioral Neuroscience and Communication (BNC), Vita-Salute San Raffaele University, 20132 Milan, Italy; 3Child and Youth Lab, Sigmund Freud University of Milan, 20143 Milan, Italy; 4Faculty of Psychology, Vita-Salute San Raffaele University, 20132 Milan, Italy

**Keywords:** delay discounting, neuromodulation, transcranial magnetic stimulation, transcranial direct current stimulation, orbitofrontal cortex, TMS, tDCS

## Abstract

In decision making, the subjective value of a reward declines with the delay to its receipt, describing a hyperbolic function. Although this phenomenon, referred to as delay discounting (DD), has been extensively characterized and reported in many animal species, still, little is known about the neuronal processes that support it. Here, after drawing a comprehensive portrait, we consider the latest neuroimaging and lesion studies, the outcomes of which often appear contradictory among comparable experimental settings. In the second part of the manuscript, we focus on a more recent and effective route of investigation: non-invasive brain stimulation (NIBS). We provide a comprehensive review of the available studies that applied transcranial magnetic stimulation (TMS) and transcranial direct current stimulation (tDCS) to affect subjects’ performance in DD tasks. The aim of our survey is not only to highlight the superiority of NIBS in investigating DD, but also to suggest targets for future experimental studies, since the regions considered in these studies represent only a fraction of the possible ones. In particular, we argue that, based on the available neurophysiological evidence from lesion and brain imaging studies, a very promising and underrepresented region for future neuromodulation studies investigating DD is the orbitofrontal cortex.

## 1. Introduction

In the last several decades, cognitive neuroscience has assisted the development of powerful tools that allow the measurement and modulation of brain activity with good spatial resolution. Such advances have made it possible to answer specific questions about the relevance of individual brain areas in behavior and cognition, although, in a holistic perspective, complex networks throughout the brain are thought to drive higher-order cognitive processes, such as decision making. Nevertheless, even at this higher order of functioning, our brain can still follow very basic rules, and these might be more rigidly dependent on the activity of less extended circuits. 

For instance, in the context of decision making, delay discounting (DD) is a phenomenon observed when subjects have to evaluate rewards of different magnitudes with delays to receipt. According to DD, the subjective value of a reward, normally measured trough intertemporal choice tasks (Figure 1a), declines with the delay to its receipt [1]. Although different models have been applied to fit the devaluation function of DD, the most established is the hyperbolic (Figure 1b, adapted from [2]). Both DD and hyperbolic devaluation have been reliably found across different species, indicating that such processes are developmentally conserved and thus of great importance for the survival of animals [3]. Even at a biopsychosocial level of analysis, DD plays a crucial role in economic decision and policy making [4]. The quantitative nature of the process also suggests that it can be treated as a psychophysical rule, even if in the domain of cognitive functioning.

As a further proof of its importance for physiological functioning, DD has been shown to be systematically compromised in several psychiatric conditions [5,6,7], and, thanks to its trans-species validity, it appears as a very promising avenue for translational research based on animal models [8].

Depending on the available neuroscientific evidence, several interpretative models for such phenomena have been proposed which are often mutually exclusive. Based on functional magnetic resonance imaging (fMRI) experiments, McClure et al. hypothesized the existence of two interacting systems that support decision making, named β and δ, the former relating to the “immediacy” nature of the decision outcome and the latter to “all decisions” [9]; Kable and Glimcher proposed the existence of a unique system widespread in the brain that computes the subjective value of rewards [10], while Ballard et al. speculate that the magnitude and the delay of a reward are evaluated by different neuronal networks [11]. On the other hand, based on non-invasive brain stimulation (NIBS) experiments, Figner et al. suggested the existence of a dedicated brain area which actively suppresses the impulsive choice, the one providing an immediate reward [12], and Nejati et al. postulated the involvement of “hot”, i.e., high-order psychological processes related to motivation and emotion, and “cold”, purely cognitive, networks in the DD task [13]. Scherbaum et al. instead argued that performance in an intertemporal choice task can be described by an attractor model, where the delayed reward choice has to overcome the “attraction energy/potential landscape” of the immediate reward in order to be selected [14].

In addition to the quantity and type of neural systems required for the production of delay discounting, the localization of these systems within the brain remains a topic of ongoing debate. It is important to acknowledge that, despite the apparent simplicity of the underlying logic behind temporal discounting, it is probable that multiple neural systems interact during the decision-making process; for example, in these tasks, the subject must perceive and interpret the stimuli that are predictive of the future rewards [15], and sensory systems have already been implicated in decision processes [16]. Hence, these involved systems might contribute themselves to the encoding, at least in part, of devaluation. Although an extreme reductionist/localist approach would not then be appropriate in this context, the available studies do not reject a priori the hypothesis that DD could be mostly dependent on the activity of a specific brain area. Nevertheless, based on the above premises, different methodological approaches might lead to the identification of very different targets. For these reasons, in this review we will discuss correlative studies, i.e., studies based on measuring brain activity during DD tasks, separately from causal studies, i.e., studies based on brain lesions and neuromodulation, which we consider more valuable in this context. 

## 2. Neural Correlates of Delay Discounting Assessed by Neuroimaging Studies

Numerous experimental approaches have been employed to investigate the neural mechanisms underlying DD. In the following, we conduct a qualitative examination of neuroimaging studies to examine the presence of any potential convergence among the brain regions identified. Voxel-based morphometry reveals that reduced white matter (WM) volume in the right prefrontal subgyral area and a higher WM volume in the right parahippocampus extending to the right hippocampus are associated with steeper DD performances [17]. Other authors found increased thickness of the bilateral medial prefrontal cortex (mPFC) and the anterior/midcingulate cortices to be related to individuals’ tendencies for less discounting during DD tasks [18]. 

Neuroimaging experiments could potentially unveil the neuronal regions mostly involved in DD, yet the outcomes appear to be contradictory among comparable experimental settings: in many instances, not only are different brain regions identified as significantly active during the task, but also, when the active areas are the same, their activities are related to different features of the DD task. In a session of fMRI experiments, for instance, Ballard et al. pointed out that increasing future reward delay negatively correlated with the activation of the left dorsolateral prefrontal cortex (dlPFC), the right posterior parietal cortex (PPC) and the left temporal-parietal junction (TPJ). The interaction of delay and magnitude negatively correlated with activation in the right inferior frontal gyrus (IFG). Interestingly, increasing future reward magnitude correlated with the activity of the mesial prefrontal cortex (MPFC), the posterior cingulate cortex and the right nucleus accumbens (NAc) [11]. The NAc is also widely studied in DD experiments using rodents: indeed, dopamine (DA) release dynamics in the NAcs of rats account for the encoding of both reward magnitude and delay [19], and modulating the activity of serotoninergic neurons of the dorsal raphe nucleus (DRN) projecting into the NAc seems to alter intertemporal choices [20].

On the contrary, McClure et al. observed the activation of the midbrain dopamine system, including part of the paralimbic cortex, for immediate rewards (the β system) [9]. In rodents, also, the activity of DA neurons in the VTA increases constantly while the mouse is waiting for the reward, and optogenetically manipulating the DA activity in this region alters the duration of delay gratification [21]. On the other hand, the lateral prefrontal and associated parietal areas were activated by all types of intertemporal choices (the δ system). In another study, McClure et al. showed that the interval of time sensible to the activation of the β system depends on the type of reward, i.e., money or food [22]. Other authors ascertained that the ventral striatum, mPFC and posterior cingulate cortex are involved in computing the subjective value of rewards and that their activities correlate proportionally with subjective value. Curiously, all three regions seem to increase their activity after subjects receive a reward or immediately before an expected reward [10].

In the comparison with risky decision-making processes, Peters and Büchel [23] found similar activity patterns to Kable and Glimcher [10]. Furthermore, the OFC and ventral striatum were active during both tasks, which supports their role in encoding the value for the subjective stimulus. Experiments involving Go/NoGo tasks in a group divided into lower and higher levels of delay of gratification showed lower recruitment of the inferior frontal gyrus and greater recruitment of the ventral striatum [24], albeit DD tasks were not directly administered in this experiment. Hare et al. found that a subregion of the left dlPFC was more active when subjects chose the delayed choice and that, before the choice was made, the functional connectivity between the dlPFC and mPFC increased, especially for most delayed rewards [25]. Functional and morphological connectivity unveils the importance of striatal connections, in particular those with the dlPFC and the amygdala: the first is associated with less discounting, while the second is associated with a steeper value [26].

In conclusion, from brain imaging studies, the areas most involved appear to be the dlPFC, mPFC, OFC, PPC and ventral striatum. Nevertheless, from these experiments we cannot infer the effective involvement of these regions due to the non-specificity of BOLD signals or kindred indirect measures. Additionally, given the wide-ranging significance of these areas in decision making and cognate processes, the observed concurrent activation may simply be a byproduct of delay discounting (DD). Working memory, for instance, is implicated in intertemporal choices task [27]. However, the relationship between this cognitive domain and DD is still elusive: aging-dependent working memory decline cannot explain the enhanced ability of delay gratification in older people [4]. Moreover, the hippocampus and basolateral amygdala are able to modulate the discounting process as well, suggesting that episodic memory circuits of the medial temporal lobe are also engaged [28].

The results of these studies indicate that the neuronal circuits utilized for delay discounting tasks are highly diverse and likely intertwined with several psychological factors, sometimes leading to conflicting findings, which prevent the development of a comprehensive model of DD functioning. To address this issue, an active intervention to differentiate specific brain regions might be very effective. In the following paragraphs, we will present and discuss causal investigations based on the manipulation of brain activity.

## 3. Investigating the Causal Role of Neural Circuits in Delay Discounting

The conventional method for investigating the causal relationships, i.e., the precise role and function, of a specific brain region in a psychological process is through the study of the consequences of brain lesions, damage or injury to that neural circuit. Unfortunately, these studies provide a highly contentious picture. Indeed, several combinations of frontal lobe injuries in humans seem not to dramatically alter performance in DD tasks [29], while lesions of the orbitofrontal cortex (OFC) increase preferences for small, immediate rewards [30]. It is worth mentioning that lesioned areas are not precisely defined for the different types of accidents that cause the lesions. Furthermore, plastic processes that follow brain injury could lead to important and heterogeneous changes in lesioned and proximal brain structures, likely causing great inter-subject variability. Several neuromodulation methods have flourished in recent years, including photobiomodulation [31] and optogenetic [32,33] and ultrasound [34] stimulation. However, at present, only a few experimental studies make use of the abovementioned techniques in humans. On the other hand, transcranial magnetic stimulation (TMS) and transcranial direct current stimulation (tDCS) represent the most-exploited and -widespread techniques in the current neuroscientific literature and have already been applied in evaluating the role of several cortical regions in DD.

### Effects of Transcranial Magnetic Stimulation (TMS) Neuromodulation on Delay Discounting

TMS, one of the main NIBS techniques, has proven successful in the treatment of many severe clinical conditions, including both psychiatric and neurological diseases [35]. For most clinical applications, the efficacy of TMS is thought to rely on the induction of synaptic plasticity and neuronal excitability in the targeted areas and circuits, which likely produces enduring changes in their activity levels [36,37,38]. In detail, low-frequency stimulation, of 1 Hz or less, causes a lasting decrease in cortical excitability, while high-frequency TMS stimulation, of 5 Hz or more, including either continuous or intermittent theta burst stimulation (TBS), has the opposite effect [39]. As for basic research, TMS represents a very valuable tool for investigating the functional roles of specific brain areas in the behavioral and cognitive functioning of human subjects. In the context of DD, the online manipulation of neuronal regions could more effectively reveal the functional role of a specific area, providing not only the opportunity to localize the phenomenon, but also the potential to decompose it into basic functional domains. This represents a crucial step towards a complete understanding of DD.

However, even using the very general search string “((delay discounting) OR (intertemporal choice) OR (impulsive choice) OR (temporal discounting)) AND (TMS)” in PubMed, only nine results were obtained at the time of writing this review. Most of the experiments described in these studies (summarized in Table 1) investigated the role of the dlPFC [12,15,40,41,42,43,44,45,46,47,48], but the results remain controversial. Based on the literature described above, one might assume that low-frequency stimulation of the dlPFC would in turn augment the discounting rate in intertemporal choice tasks, while high-frequency stimulations, on the contrary, would diminish it. Indeed, high-frequency TMS, when applied to the dlPFC, reduces DD in smokers, but not the daily amount of cigarettes they smoke [42], while reducing the activity of the dlPFC with low-frequency stimulation increased discounting [46]. Nevertheless, 20 Hz TMS applied to the dlPFCs of subjects with depression did not affect DD performance [40]. In a comparison of neuromodulation of the left and right hemispheres, we also found some conflictual outcomes in the available literature: only low-frequency stimulation of the left dlPFC causes delay discounting rates to decrease [12,46]. 

Two studies investigated the role of the mPFC using high-frequency stimulation, leading to divergent findings: Zack et al. found that the stimulation had no effect [41], while Cho et al. obtained a decrease in DD [49]. In another study, continuous theta burst stimulation was applied to the posterior temporal-parietal junction (pTPJ), leading to an increase in delay discounting [50].

In the previous section, the OFC and ventral striatum were indicated as potentially involved regions, but, unfortunately, targeting these brain areas with TMS is rather challenging due to their locations: while the ventral striatum is too deep to be efficiently reached by the magnetic field, OFC stimulation, albeit practicable, is painful for the subject due to the unwanted contraction of facial muscles [51]. Therefore, we can argue that extensive investigation of the dlPFC compared to the OFC might be related to physical constraints rather than evidence-based hypotheses.

The absence of information regarding the impacts of stimulation on other relevant brain regions during delay discounting tasks constrains our ability to comprehend whether such regions play a direct or indirect role in the process. In addition, the inclusion of patients suffering from psychiatric disorders in some experiments [40,41,42,44] could introduce artifacts.

**Table 1 brainsci-13-00403-t001:** Studies investigating delay discounting performance modulation by TMS. MDD = major depression disorder; PG = pathological gambling; AUD = alcohol use disorder; MT = motor threshold.

Study	Brain Area	Protocol	Stimulation Intensity	n	Disease	Discounting Assessment	Effect
Cho et al., 2015 [51]	mPFC	10 Hz-rTMS	80% MT	24	-	ln (k)	Decreased delay discounting
Zack et al., 2016 [40]	mPFC	rTMS	80% MT	9	PG	k	-
Zack et al., 2016 [40]	Right dlPFC	TBS	80% MT	9	PG	k	-
Cho et al., 2010 [42]	Right dlPFC	cTBS	80% MT	7	-	k	Decreased delay discounting
Figner et al., 2010 [12]	Right dlPFC	1 Hz low-frequency rTMS	54% MT	19	-	Immediate choices (%)	-
Cho et al., 2012 [46]	Right dlPFC	cTBS	80% MT	8	-	ln (k)	Decreased delay discounting
Schluter et al., 2019 [43]	Right dlPFC	HF-rTMS	110% MT	40	AUD	AUC	-
Ballard et al., 2018 [45]	Right dlPFC	1 Hz low-frequency rTMS	120% MT	12	-	log (k)	Increased delay discounting
Essex et al., 2012 [44]	Right dlPFC + right PPC	1 Hz low-frequency rTMS	54% MT	16	-	Immediate choices (%)	Decreased delay discounting
Teti Mayer et al., 2019 [39]	Left dlPFC	10 Hz	110% MT	20	MDD	k	-
Sheffer et al., 2013 [41]	Left dlPFC	HF rTMS	110% MT	47	Smokers	k	Decreased delay discounting
Figner et al., 2010 [12]	Left dlPFC	1 Hz low-frequency rTMS	54% MT	18	-	Immediate choices (%)	Increased delay discounting
Ballard et al., 2018 [45]	Left dlPFC	1 Hz low-frequency rTMS	120% MT	15	-	log (k)	Increased delay discounting
Yang et al., 2018 [52]	Left dlPFC	iTBS	80% MT	23	-	ln (k)	-
Essex et al., 2012 [44]	Left dlPFC + left PPC	1 Hz low-frequency rTMS	54% MT	16	-	Immediate choices (%)	Increased delay discounting
Soutschek et al., 2016 [49]	Right pTPJ	cTBS	80% MT	22	-	log (k)	Increased delay discounting
Soutschek et al., 2016 [49]	Right pTPJ	cTBS	80% MT	20	-	log (k)	Increased delay discounting
Soutschek et al., 2016 [49]	Left S1	cTBS	80% MT	21	-	log (k)	-

## 4. Effects of Transcranial Direct Current Stimulation (tDCS) Neuromodulation on Delay Discounting

Among non-invasive brain stimulation techniques, tDCS could be a cheaper and portable alternative to TMS. When one applies a positive current flow through one electrode, i.e., anodal stimulation, the excitability of the cortical neurons under that area of the scalp is increased, while a negative current (cathodal) decreases their excitability. Therefore, tDCS is thought to facilitate or inhibit the electrical activation of targeted brain regions [52].

Brunyé reviewed the latest research on the effects of non-invasive brain stimulation on decision-making processes and found that some protocols of tDCS can effectively modulate performance in several domains, such as economic, risky and perceptual decision making, as well as moral tasks [53].

Searching for tDCS applications in the field of DD, we entered the string “((delay discounting) OR (intertemporal choice) OR (impulsive choice) OR (temporal discounting)) AND (tDCS)” and were able to find a greater number of studies compared to the TMS case, although the figure was still as low as 20 (summarized in Table 2). 

Most of the experiments investigated the role of dlPFC. Specifically, the anodal stimulation of the left dlPFC appears to reduce discounting rates in most trials [13,54,55,56,57,58]; nevertheless, some authors have found no significant effect of stimulation [59,60,61]. It is worth mentioning that the cathodal stimulation of this brain area strongly correlates with an increase in discounting rates [55,59]. The vmPFC also appears as a promising area, since its anodal stimulation leads to a reduction in the steepness of the DD curve [60,62]. The role of the OFC has been the focus of a study using tDCS: both anodal and cathodal stimulation at frontal sites caused a decrease in the discounting of delayed rewards [13]. Nevertheless, after performing a simulation based on SimNIBS (simnibs.github.io) of the electrical field produced by the electrodes’ configuration used in this study, we observed the current flowing in most of the PFC subregions, with only marginal, if any, involvement of the OFC (data not shown).

Finally, the stimulation of the cerebellum [63], the inferior frontal gyrus (IFG; [64]) and the motor cortex (M1; [61]) has been found to cause no alteration in intertemporal choice task performance.

One issue with tDCS relates to the fact that stimulation is not well-confined to specific brain regions, as compared to TMS, not only due to the dimensions of the electrodes but also depending on the placement of the reference electrode [65]. Even though high-density tDCS can be an effective choice for increasing spatial confinement, the use of computational modeling of current flows can also support the better design of experiments targeting specific regions, including subcortical ones [66]. As for TMS, the plethora of studies investigating DD with tDCS cannot provide us with enough information for understanding the neuronal circuits that are involved in this phenomenon. As a matter of fact, even with tDCS, there is a disproportionate number of studies investigating the dlPFC, although the OFC and ventromedial prefrontal cortex (vmPFC) clearly appear as promising targets.

**Table 2 brainsci-13-00403-t002:** Studies investigating delay discounting performance modulation by tDCS. PD = Parkinson’s disease; ADHD = attention deficit hyperactivity disorder; BN = bulimia nervosa; CUD = cocaine use disorder.

Study	Anode Position	Cathode Position	Current Intensity	n	Disease	Discounting Assessment	Effect
Manuel at al., 2019 [63]	vmPFC	Vertex	2.0 mA	20	-	log (k)	Decreased delay discounting
Nejati et al., 2021 [61]	Right vmPFC	Left dlPFC	1.0 mA	20	ADHD children	k	Decreased delay discounting
Wang et al., 2021 [67]	FPC	Vertex	1.5 mA	90	-	k	-
Soutschek et al., 2017 [68]	Left FPC	Vertex	1.0 mA	27	-	Immediate choices (%)	-
To et al., 2018 [69]	Right IFG	Left IFG	2.0 mA	23	Chocolate cravers	k	-
Nejati et al., 2018 [13]	Right OFC	Left dlPFC	1.5 mA	24	-	k	Decreased delay discounting
He at al., 2016 [55]	Right dlPFC	-	1.5 mA	23	-	k	-
Shen et al., 2016 [56]	Right dlPFC	Left dlPFC	2.0 mA	39	-	k	-
Shen et al., 2016 [56]	Right dlPFC	-	2.0 mA	39	-	k	-
Xiong et al., 2019 [58]	Right dlPFC	Left dlPFC	1.5 mA	20	-	k	-
Kekic et al., 2017 [57]	Right dlPFC	Left dlPFC	2.0 mA	39	BN	δ	Decreased delay discounting
Kekic et al., 2014 [70]	Right dlPFC	Left dlPFC	2.0 mA	17	Food cravers	k	-
Hecht et al., 2013 [71]	Right dlPFC	Left dlPFC	1.6 mA	14	-	Immediate choices (%)	Decreased delay discounting
Brunelin and Fecteau, 2021 [59]	Left dlPFC	Right dlPFC	2.0 mA	15	Acutely stressed	Immediate choices (%)	Decreased delay discounting
He at al., 2016 [55]	Left dlPFC	-	1.5 mA	22	-	k	Decreased delay discounting
Shen et al., 2016 [56]	Left dlPFC	Right dlPFC	2.0 mA	39	-	k	-
Shen et al., 2016 [56]	Left dlPFC	-	2.0 mA	39	-	k	Decreased delay discounting
Nejati et al., 2018 [13]	Left dlPFC	Right OFC	1.5 mA	24	-	k	Decreased delay discounting
Terenzi et al., 2021 [62]	Left dlPFC	Right shoulder	1.5 mA	28	PD	log (k)	-
Nejati et al., 2021 [61]	Left dlPFC	Right vmPFC	1.0 mA	20	ADHD children	k	-
Xiong et al., 2019 [58]	Left dlPFC	Right dlPFC	2.0 mA	20	-	k	Decreased delay discounting
Kekic et al., 2017 [57]	Left dlPFC	Right dlPFC	2.0 mA	39	BN	δ	Decreased delay discounting
Gaudreault et al., 2021 [72]	Left dlPFC	Right dlPFC	2.0 mA	17	CUD	k	Decreased delay discounting
Colombo et al., 2020 [60]	Left dlPFC	Right triceps	1.5 mA	13	-	RT	-
Hecht et al., 2013 [71]	Left dlPFC	Right dlPFC	1.6 mA	14	-	Immediate choices (%)	Decreased delay discounting
Manuel at al., 2019 [63]	Vertex	vmPFC	2.0 mA	20	-	log (k)	-
Wang et al., 2021 [67]	Vertex	FPC	1.5 mA	90	-	k	-
Soutschek et al., 2017 [68]	Vertex	Left FPC	1.0 mA	26	-	Immediate choices (%)	-
Terenzi et al., 2021 [62]	Left M1	Right shoulder	1.5 mA	28	PD	log (k)	-
Wynn et al., 2019 [64]	Medial cerebellum	Right deltoid muscle	2.0 mA	26	-	AUC	-
Colombo et al., 2020 [60]	Right triceps	Left dlPFC	1.5 mA	13	-	RT	Increased delay discounting
Shen et al., 2016 [56]	-	Right dlPFC	2.0 mA	39	-	k	-
Shen et al., 2016 [56]	-	Left dlPFC	2.0 mA	39	-	k	Increased delay discounting

## 5. Conclusions, Limitations and Future Developments

Although the results of neuroimaging studies do not provide a definitive understanding of the specific contributions of brain regions to delay discounting tasks, they are valuable in identifying potential targets for investigation using non-invasive brain stimulation techniques. To synthesize the current state of research in this field, we present a schematic brain map (Figure 2) highlighting the brain regions implicated in delay discounting based on both neuroimaging studies (green areas) and neuromodulation studies (red areas), with yellow areas indicating the overlap between the two. If we consider the main hypothesis that DD is processed in a unique area, the left dlPFC appears to be the best candidate: excitatory stimulation of this area produces a decrease in discounting [13,42,54,55,56,57,58], while, on the contrary, its inhibition produces a steeper DD characteristic [12,46,55,59].

Nevertheless, there exist several caveats that might limit the significance of the results. In most of the studies evaluated here, for instance, the rewards were monetary (e.g., [13,55,57]), but, even if the general hyperbolic function applies, one cannot exclude that primary rewards (i.e., food [64,67] and sexually-related images [68]) are processed by distinct neural circuits [22]. The timing of rewards, moreover, is very heterogeneous among the experimental settings, and this could potentially result in the recruitment of completely different neuronal circuits [15]. Indeed, the left caudate nucleus, the ventral striatum and the putamen show significantly greater activation during trials in which the hypothetical rewards are associated with delays shorter than one year compared to those with delays longer than one year [69]. Finally, other confounds are introduced by the compositions of experimental groups, some of which include patients with psychiatric and/or neurological conditions.

Furthermore, investigating the roles of areas other than the dlPFC seems very promising, these areas being underrepresented. OFC activity, above all, seems to be engaged in DD, both in humans [9,10,22,23,30,70] and animals. Indeed, the optogenetical stimulation of serotoninergic terminals of the OFC, rather than the NAc and mPFC, promotes waiting for behavior in intertemporal choice tasks [71]. Being highly interconnected with the VTA through the NAc, dlPFC, DRN, amygdala and hippocampus [72], the OFC can easily access all the information required for the expression of functional DD and at the same time it can exert top-down control on areas involved in the encoding of reward values. Specifically, the medial orbitofrontal cortex (OFC) is a component of the corticostriatal circuit implicated in the assessment of the subjective value of rewards and is thus likely to be involved in delay discounting [72].

Furthermore, the OFC is particularly vulnerable to age-related reduction in gray matter volume [74], possibly explaining why older people prefer delayed rather than immediate rewards. Finally, the WM of the OFC is altered in patients with anorexia nervosa, who, contrary to most of the psychiatric population, show a preference for most delayed rewards [75]. Unfortunately, only a few studies have investigated the effects of non-invasive neuromodulation of this region on DD (Figure 2).

Nevertheless, based on the available evidence, one could hardly formulate a theoretical model that describes the functioning of DD, since intertemporal choice tasks are likely to involve the interaction of complex systems comprising several brain regions. Hence, we believe that future studies should not only better elucidate the role of the OFC in this context, but also address the potential routes of its interaction with other brain areas known to be pivotal in decision making.

A potential caveat could arise from the poor comprehension of the effect of TMS and tDCS protocols on targeted neuronal circuits. For instance, there is a lack of consensus among authors regarding the extent to which certain TMS protocols can simulate a “virtual lesion” [76]. Additionally, in the reviewed experiments, delay discounting is evaluated either in real time, i.e., during the administration of the stimulation protocol, or after prolonged exposure to treatment.

The active manipulation of brain regions offers a more comprehensive means of dissecting and comprehending delay discounting than a solely descriptive approach, such as neuroimaging. Furthermore, given that delay discounting is compromised in a number of treatment-resistant psychiatric conditions, such as addiction [7], the utilization of non-invasive brain stimulation techniques may hold potential for clinical applications, as the rehabilitation of this psychological function may result in improved patient outcomes.

## Figures and Tables

**Figure 1 brainsci-13-00403-f001:**
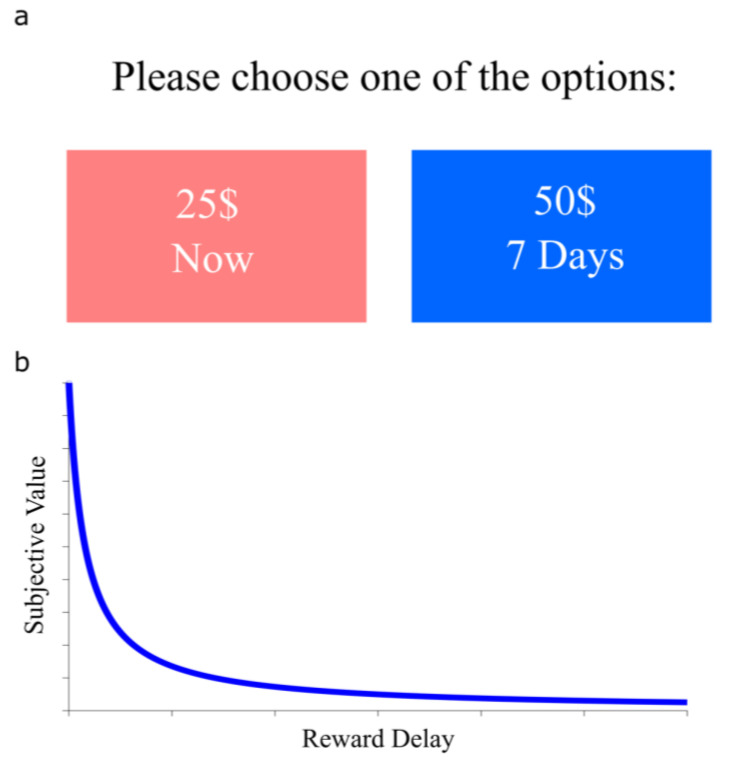
Intertemporal choice tasks for characterizing delay discounting in decision making. (**a**) An exemplary question from an intertemporal choice task. (**b**) The hyperbolic discounting function fitted from the indifference points obtained from the intertemporal choice task.

**Figure 2 brainsci-13-00403-f002:**
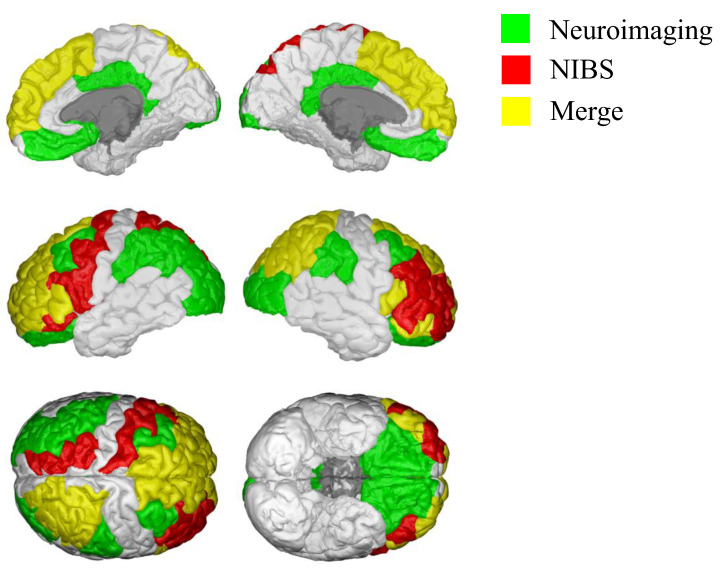
Cortical areas involved in delay discounting. A qualitative reconstruction was obtained using the Desikan–Killiany cortical atlas and BrainPainter [73] to summarize the areas involved in delay discounting determined by neuroimaging studies (green areas) and non-invasive brain imaging (NIBS; red areas); yellow areas indicate the overlap between the two.

## Data Availability

Data sharing is not applicable to this article, as no new data were created or analyzed in this study.

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
