# Peer review of "Neural Correlates of Delay Discounting in the Light of Brain Imaging and Non-Invasive Brain Stimulation: What We Know and What Is Missed"

_brainsci, 2023, doi:10.3390/brainsci13030403_

Round 1
Author Response
This review summarizes recent research progress on the topic of delay discounting using neuroimaging and/or non-invasive brain stimulation methods. It is well conducted, covers significant contributions on this topic and importantly points out what has been missed in those studies. However, the organization and writing of the review can be significantly improved before it can be considered to be published.
We greatly thank the reviewer for the careful evaluation of our manuscript and for the positive comments about its contributions. We hope that the following replies to reviewer’s concerns, together with the revised version of the manuscript, will address all raised issues.
- Line 80: here the “temporal devaluation” term is used for the first time (also seem to be the only time) without being explained what it means. In the delay discounting field, “temporal discounting” is a commonly-adopted term to describe the mechanism that the subjective value of a future reward declines with delays. Does the term “temporal devaluation” mean the same thing or slightly different?
Thanks for the indication, we replaced the term with a more established one for sake of clarity.
- Line 88-91: The reasoning for grouping lesion studies and fMRI studies and separating them from neuromodulation studies are not convincing to me. Typically, the lesion studies and neuromodulation studies are thought to be analogous in terms of providing a causal intervention. Of course, as the authors mentioned in lines 107-110, the lesion studies are not without limitations, including that the lesion regions may not be precisely defined, and the plastic process following brain injury, but these limitations cannot deny the causal nature of lesion studies. The authors could either (1) reorganize the review based on correlational (fMRI) and causal (lesion, TMS, tDCS) studies, or (2) carefully and specifically justify why the current organization is reasonable before separately discussing each.
The reviewer is right since the paragraph about lesion studies is not well positioned in the main text. Hence, we reorganized the manuscript so that only neuroimaging studies are discussed in chapter 2, while in chapter 3 we describe all causal studies, including lesion studies (in the introductory paragraph), TMS studies in section 3.1 and tDCS studies in section 3.2. We believe that such solution greatly enhances the structure of the review. We thank the reviewer for this suggestion.
- A general comment on discussing specific studies. When possible, it helps to mention what kind of populations were used, what type of rewards were used, and any other information unique to that study.
We included this information as indicated by the reviewer. However, most studies use financial rewards. We listed the other important details of the experimental studies in the tables for sake of clarity. We also added a final discussion about such heterogeneity in the conclusions of the revised manuscript. Thanks for the indication.
- Line 128: “this latter area”. Please be specific.
Done.
- Line 138: “a study by the same group”. Please be specific about which group was referred to.
Done.
- Line 156-157: although this is a conclusion, it’s too ambiguous to say these regions are simply “involved”. A clearer conclusion sentence can summarize the most common findings or their different directions of effect on DD (i.e. promoting delayed choices or immediate choices).
This is a very crucial point. Unfortunately, from these correlation studies it is not possible to deduce a specific function of the involved circuit in promoting either delayed or immediate choices (mainly for constraints in temporal resolution): this is one of the most important limitations of such an approach, as we discuss in the manuscript. Even if an experimental design to address this question were possible, it has not been evaluated to our knowledge. We now point out such aspect in the conclusions of the manuscript. Thanks.
- Line 165-171: the reasoning for why NIBS studies are needed is not convincing. If the goal is to “build a comprehensive model of DD functioning”, why “intervening specific brain parts” would be helpful? Did the authors propose here that different NIBS studies targeting different brain areas should be conducted? Please clarify and improve the reasoning in here.
The reviewer is right as the reasoning is not clear. We rephrased the comment to explain why such causal studies can be very valuable to understand neural mechanisms behind DD. We thank the reviewer for this important indication.
Line 196: what does “but not the consumption of cigarettes” mean here? Was that referred to this population or their behavior on cigarettes?
We better specified that experimental investigation in the revised manuscript.
- Line 199-201: The authors listed multiple TMS-dlPFC studies on DD in line 192, but only cited 12 and 40 when discussing the left vs. right side of TMS. What happened in the other studies? Did they report this information at all? Why were they not discussed here?
We are sorry for this lack of clarity. The reason was that those studies are the only one providing coherent results in terms of stimulation sites, hemisphere, and effect’s direction. The other ones either did not find any significant effect or provide contradictory results. We better state this reasoning in the revised version of the paper.
- Line 202-203: what kind of opposite findings from mPFC? Were they opposite from the findings from dlPFC, or opposite between themselves?
Thanks for raising this issue, we better explained this aspect in the revised manuscript.
- Line 203-204: given that the effects of low/high frequency of TMS was explained in line 180-181, the effect of cTBS/iTBS should also be explained to be clear. What does the word “appears to” mean?
The reviewer is right about this missing. We now included the explanation about TBS protocols and rephrased the sentence.
- Line 206-212: given that TMS on OFC studies does exist, it’s not proper to list it as “not feasible”.
Thanks, we rephrased the sentence.
- Line 245-250: why “as stated by the authors” highlighted here? Was the discounting indeed decreased by both anodal and cathodal stimulation? The simulation by the SimNIBS sounds interesting, but it cannot be used as an argument without the results reported already.
We are sorry for this lack of clarity. We rephrased the sentence to better explain our viewpoint. As the matter of fact, both stimulus configurations produce a decrease in discounting. Nevertheless, the authors’ comment about the implicated areas should be treated as an inference due to tDCS limitations, and we simply questioned the robustness of such inference based on simulation data. We clarified our viewpoint in the revised manuscript. Thanks for the indication.
- Line 253-255: this argument requires proper references.
Done.
- Fig 2: it’s hard to construe the figure without talking about how each region contributes to DD.
Unfortunately, we believe that, due to results’ heterogeneity, it would be very confounding to show specific aspects in the image. Hence, the aim of this figure is only to summarize the implicated/investigated areas in terms of localization, to highlight the degree of superimposition. We believe that this might be of interest also to identify more promising targets or underestimated ones.
Some minor issues/typos
- Line 102: qualitatively
Corrected
- Line 159: “by be indirectly”?
Corrected
- Line 160: “expression of delay discounting”. It’s hard to understand “expression” here.
Corrected
- Fig 2: it’s better to use more intuitive color combinations to indicate the overlap. For example, the overlap between red and green is yellow.
We perfectly agree, hence we inverted colors using yellow to indicate merge. Thanks for suggestion.
We heartly thank the reviewer for the in-depth analysis of our paper and for all the suggestions. We believe that these really improved the clarity and impact of our manuscript.
Reviewer 2 Report
At the manuscript "Neural correlates of delay discounting in the light of brain imaging and non-invasive brain stimulation: what we know and what is missed" by Drs. Andrea Stefano Moro et al authors considered the latest neuroimaging and clinical studies related with delay discounting (DD). Also authors reviewed recent achievements in the field of non-invasive brain stimulation (NIBS) including transcranial magnetic stimulation (TMS) and transcranial direct current stimulation (tDCS), affecting the subject’s performance in DD tasks.
Impressive manuscript, authors have touched upon an extremely important and complex topic, and it is certain that this manuscript will resonate with the scientific community.
I have only minor criticisms:
As the authors rightly write, being one of the main noninvasive brain stimulation techniques Transcranial Magnetic Stimulation (TMS) has proven successful for the treatment of many clinical conditions. However, many aspects of the physiological basis of the effect of TMS on the brain are still debatable. In the topic that the authors are investigating, the role of the physiological basis of the action of TMS is very large. I would like to know the opinion of the authors, and advise to use some publications on this topic. Including:
Garcia-Sanz et al. Use of transcranial magnetic stimulation for studying the neural basis of numerical cognition: A systematic review. J Neurosci Methods. 2022 1;369:109485. doi: 10.1016/j.jneumeth.2022.109485
Sven Bestmann; The physiological basis of transcranial magnetic stimulation; Affiliations expand; PMID: 18243042 DOI: 10.1016/j.tics.2007.12.002
There is also a more theoretical question - what are the prospects for using modern non-invasive (or minimally invasive) methods of brain stimulation in clinical practice? Hybrid technologies such as:
Chen et al; Wireless Optogenetic Modulation of Cortical Neurons Enabled by Radioluminescent Nanoparticles. ACS Nano. 2021 23;15(3):5201-5208. doi: 10.1021/acsnano.0c10436.
Can ultrasonic be used? Such as:
Darmani et al. Non-invasive transcranial ultrasound stimulation for neuromodulation. Clin Neurophysiol. 2022 Mar;135:51-73. doi: 10.1016/j.clinph.2021.12.010.
Authors should briefly highlight this side of the issue by citing these (and possibly some others) publications. The presentation of a subject is systematic and comprehensive. I am happy to recommend the manuscript for the publication after minor corrections mentioned above.
Author Response
At the manuscript "Neural correlates of delay discounting in the light of brain imaging and non-invasive brain stimulation: what we know and what is missed" by Drs. Andrea Stefano Moro et al authors considered the latest neuroimaging and clinical studies related with delay discounting (DD). Also authors reviewed recent achievements in the field of non-invasive brain stimulation (NIBS) including transcranial magnetic stimulation (TMS) and transcranial direct current stimulation (tDCS), affecting the subject’s performance in DD tasks.
Impressive manuscript, authors have touched upon an extremely important and complex topic, and it is certain that this manuscript will resonate with the scientific community.
We heartily thank the reviewer for the very positive comments about our manuscript and for the useful suggestions. We hope that the revised version of the manuscript will be appreciated.
I have only minor criticisms:
As the authors rightly write, being one of the main noninvasive brain stimulation techniques Transcranial Magnetic Stimulation (TMS) has proven successful for the treatment of many clinical conditions. However, many aspects of the physiological basis of the effect of TMS on the brain are still debatable. In the topic that the authors are investigating, the role of the physiological basis of the action of TMS is very large. I would like to know the opinion of the authors, and advise to use some publications on this topic. Including:
Garcia-Sanz et al. Use of transcranial magnetic stimulation for studying the neural basis of numerical cognition: A systematic review. J Neurosci Methods. 2022 1;369:109485. doi: 10.1016/j.jneumeth.2022.109485
Sven Bestmann; The physiological basis of transcranial magnetic stimulation; Affiliations expand; PMID: 18243042 DOI: 10.1016/j.tics.2007.12.002
Thanks for the indications, we included our comments on the physiological bases on TMS effects in the revised manuscript, also including the suggested references. We believe that this is an important point and we previously addressed it in a previous paper (Ferro, M., Lamanna, J., Spadini, S. et al. Synaptic plasticity mechanisms behind TMS efficacy: insights from its application to animal models. J Neural Transm 129, 25–36 (2022). https://doi.org/10.1007/s00702-021-02436-7).
There is also a more theoretical question - what are the prospects for using modern non-invasive (or minimally invasive) methods of brain stimulation in clinical practice? Hybrid technologies such as:
Chen et al; Wireless Optogenetic Modulation of Cortical Neurons Enabled by Radioluminescent Nanoparticles. ACS Nano. 2021 23;15(3):5201-5208. doi: 10.1021/acsnano.0c10436.
Can ultrasonic be used? Such as:
Darmani et al. Non-invasive transcranial ultrasound stimulation for neuromodulation. Clin Neurophysiol. 2022 Mar;135:51-73. doi: 10.1016/j.clinph.2021.12.010.
This is a very interesting point, we added a paragraph about the future potential application on recently developed techniques such as those mentioned by the reviewer. Thanks for the suggestion.
Authors should briefly highlight this side of the issue by citing these (and possibly some others) publications. The presentation of a subject is systematic and comprehensive. I am happy to recommend the manuscript for the publication after minor corrections mentioned above.
We greatly thank the reviewer for the careful evaluation of our manuscript and for the recommendation.
Round 2
Reviewer 1 Report
The authors have carefully addressed my questions and concerns in this revised version.